# Oncogenic Pathways and Targeted Therapies in Ovarian Cancer

**DOI:** 10.3390/biom14050585

**Published:** 2024-05-15

**Authors:** Carolina Lliberos, Gary Richardson, Antonella Papa

**Affiliations:** 1Cancer Program, Monash Biomedicine Discovery Institute, Department of Biochemistry and Molecular Biology, Monash University, Clayton, VIC 3800, Australia; carolina.lliberos@monash.edu; 2Neil Beauglehall Department of Medical Oncology Research, Cabrini Health, Malvern, VIC 3144, Australia

**Keywords:** ovarian cancer, signalling pathways, targeted therapies, PI3K, MAPK

## Abstract

Epithelial ovarian cancer (EOC) is one of the most aggressive forms of gynaecological malignancies. Survival rates for women diagnosed with OC remain poor as most patients are diagnosed with advanced disease. Debulking surgery and platinum-based therapies are the current mainstay for OC treatment. However, and despite achieving initial remission, a significant portion of patients will relapse because of innate and acquired resistance, at which point the disease is considered incurable. In view of this, novel detection strategies and therapeutic approaches are needed to improve outcomes and survival of OC patients. In this review, we summarize our current knowledge of the genetic landscape and molecular pathways underpinning OC and its many subtypes. By examining therapeutic strategies explored in preclinical and clinical settings, we highlight the importance of decoding how single and convergent genetic alterations co-exist and drive OC progression and resistance to current treatments. We also propose that core signalling pathways such as the PI3K and MAPK pathways play critical roles in the origin of diverse OC subtypes and can become new targets in combination with known DNA damage repair pathways for the development of tailored and more effective anti-cancer treatments.

## 1. Introduction

In 2022 there were ~325,000 new reported cases of ovarian cancer (OC) and ~207,000 OC deaths worldwide [1]. By 2045, the global incidence of OC is expected to increase by 47%, with a projected rise in deaths of 58% [2]. While OC is the 11th most common cancer among women (2.5% of cancers), it is the 5th leading cause of cancer-related death, with a relative five-year survival rate of ~50.8% [3].

Only 15% of OC cases are identified during the early and localized stage of the disease, which is generally associated with better outcomes and a 5-year survival rate of ~90% [4]. However, a large portion of OC, ~60%, are diagnosed when the cancer has already metastasized to distant sites, a condition that significantly reduces the 5-year survival rate to ~30% [4]. These numbers highlight the urgent need to identify new OC biomarkers for earlier detection and more rapid intervention.

Current screening methods for OC detection are based on pelvic examination, including transvaginal ultrasound and evaluation of the cancer antigen 125 (CA-125), a peptide produced by OC cells together with mesothelial cells from multiple tissues such as the endometrium [4,5]. Once diagnosed, standard first-line treatment for OC includes surgery followed by platinum-based chemotherapy. Although 75–80% of OC patients respond to these initial treatments, ~70% will eventually relapse and develop resistance [6]. In an effort to improve patients survival and overcome resistance to standard therapies, extensive research is underway to better understand the mechanisms driving OC formation and progression, with a view to discover and improve efficacy of targeting oncogenic pathways [7]. Currently, OC remains the most lethal cancer among all gynaecological malignancies [7].

A few OC risk factors have been identified so far and include genetic predispositions and environmental issues, as well as lifestyle factors such as cigarette smoking, diet, and hormone replacement therapies [8,9]. Inherited gene mutations in the tumour suppressors *breast cancer 1* (*BRCA1*) and *breast cancer 2* (*BRCA2*) have been strongly associated with predisposition to OC. Age is also considered an important risk factor, with increased age and menopause defining a higher risk cohort of patients [8,9]. However, while these factors play a contributing role, the critical molecular mechanisms underpinning OC development and heterogeneity are complex and not fully understood.

In this review, we will summarize our current understanding of the mechanisms causing OC initiation and progression and will highlight important signalling pathways and gene mutations whose contribution to OC have not been fully explored and that can inform new and improved treatments.

## 2. Ovarian Cancer: Pathological Classification, Subtyping and Associated Mutations

The ovary is responsible for producing fully mature and developmentally competent oocytes for fertilization, and to generate hormones and growth factors that are essential for ovarian function, pregnancy, and female development [10]. The ovary has two major compartments which differ in their composition and structure: the outer cortex and the inner medulla [11]. While the ovarian cortex consists of stiff connective tissue that contains mostly quiescent ovarian follicles, the medulla is a loose connective tissue with abundant blood and lymphatic vessels harbouring growing follicles [11] (Figure 1).

OC can originate from various cell types within the ovary, including cells from the ovarian surface epithelium (OSE), germ cells, stromal cells such as fibroblasts, and other less common cell types such as mesothelial–mesenchymal cells [12]. The predominant subtype of OC comprises epithelial tumours from the OSE, which accounts for approximately 90% of cases [12,13].

Based on morphological, immunohistochemical, and genetic studies, epithelial ovarian cancer (EOC) can be further divided into two subgroups: (1) type I tumours, which include endometrioid, clear cell, mucinous, and low-grade serous carcinoma; and (2) type II, including high-grade serous carcinoma, carcinosarcomas, and undifferentiated carcinomas [14] (Figure 2).

### 2.1. Type I EOCs

Type I EOCs account for 28% of OC cases and are typically discovered through pelvic examination when the tumour is still localized in the ovary, and surgical removal allows for higher curative rates compared to type II EOCs [15,16]. These tumours tend to be of low grade, are genetically stable, and rarely acquire mutations in the *tumour protein p53* (*TP53*) a common genetic event in OC [12]. Despite this, each morphological subtype within this category is highly heterogeneous and presents a unique molecular profile defined by the activation of specific cell signalling pathways.

#### 2.1.1. Clear Cell and Endometrioid Ovarian Cancer

Ovarian clear cell carcinoma (CCC) and endometrioid ovarian cancer (EnOC) are associated with endometriosis and are diagnosed in ~10% of OC cases [17,18]. Despite being classified as a type I tumour, CCCs acquire mutations and display phenotypes that position it as an intermediate category between type I and type II tumours [18]. CCCs exhibit a high prevalence of inactivating mutations in the tumour suppressor and chromatin remodelling gene *AT-rich interaction domain 1A* (*ARID1A*), which occur in approximately 50% of CCCs [19,20]. Activating mutations in the *phosphatidylinositol-4,5-bisphosphate 3-kinase catalytic alpha subunit* (*PIK3CA*) encoding PI3Kα, occur in ~40% of CCC cases [20,21]. In addition, amplification of the *erb-b2 receptor tyrosine kinase 2* (*ERBB2*) is observed in 14% of CCCs [22], while *c-mesenchymal–epithelial transition factor* (c-*Met*), encoding a tyrosine kinase receptor that activates a wide range of different cellular signalling pathways, including the MAPK and PI3K pathways [23], is amplified in 40% of CCC cases [24].

In EnOC, mutations in the β-catenin gene, *CTNNB1*, are observed in approximately 20% of cases [25]. Similar to CCC, somatic mutations in *PIK3CA* and the tumour suppressor *phosphatase and tensin homolog* (*PTEN*) have been reported in 12% and 31% of EnOC, respectively [26], along with inactivating ARID1A mutations [27], demonstrating that CCC and EnOC share common genetic events. Additionally, the genetic landscape of EnOC reveals that while less than 7% of cases harbour activating mutations in the proto-oncogene *KRAS*, *BRAF* mutations have been detected in 24% of EnOCs [28].

Through the generation of genetically engineered mouse models, it was found that the sole loss of ARID1A or PTEN in the OSE, is not sufficient to initiate ovarian tumour formation [29]. However, 59.1% of mice with *ARID1A* and *PTEN* double knock-out developed EnOC, underscoring a synergistic tumorigenic effect between these two tumour suppressors [29]. Surprisingly, concurrent loss of *ARID1A* with active PI3K in the OSE induced formation of CCC rather than EnOC [30]. While these findings emphasize the importance of PI3K pathway activation in the context of CCC and EnOC formation, they also demonstrate how mutations belonging to the same linear cascade can eventually give rise to distinct OC subtypes. Thus, a better understanding of the specific genetic events occurring in OC can lead to more effective therapies tailored to each tumour subtype.

#### 2.1.2. Mucinous Ovarian Cancer

Mucinous ovarian cancer (MOC) is a rare histological subtype of EOC representing ~3% of these malignancies [31]. MOC originates from epithelial cells lining the OSE producing mucin [32], a class of glycoproteins involved in lubrication as well as renewal and differentiation of the epithelium [33]. MOCs tend to be asymptomatic, and 80% of cases present as metastatic disease invading the gastrointestinal tract [31].

The most common genetic alterations found in MOCs occur in the MAPK pathway and include *KRAS* mutations (65%) and *ERBB2* amplification (18%) [34,35]. PI3K pathway alterations are less commonly found in MOC than CCC and EnOC (*PIK3CA* in 14% and *PTEN* in 3%) [34,36] but notably, ~60% of MOCs harboured alterations in *TP53,* even though these are more typically associate with type II EOCs [34,35].

Aberrant WNT pathway activation through *CTNNB1* mutations has been documented in 5% of MOCs [34]. In addition, inactivating mutations in the *RNF43*, a zinc finger-dependent E3 ubiquitin protein ligase that negatively regulates WNT pathway, were observed in 21% MOC [37]. These findings suggest that RNF43 may act as an important tumour suppressor in OC and that WNT pathway activation is a central player in ovarian tumours of mucinous histology.

Mutations in the *cyclin dependent kinase inhibitor 2A* (*CDKN2A*) and *ARID1A* genes were originally reported in 19% and 9% of MOCs, respectively [34,38]. However, a more recent study by Cheasley et al. used a larger cohort of primary mucinous ovarian tumours (n = 134) and reported that *CDKN2A* alterations occurred in 76% of MOC patients [35]. This significant difference indicates how additional genetic and molecular screening on larger cohorts of rare OC subtypes can still reveal important and potentially druggable oncogenic drivers.

#### 2.1.3. Low-Grade Serous Ovarian Cancer

Low-grade serous ovarian cancer (LGSOC) represents <2% of all ovarian cancers [39]. Recent studies have shed new light on the origin of this subtype of OC, showing that LGSOC is more likely to arise from the fallopian tube epithelium (FTE) rather than from the OSE, as previously believed [40]. It has been hypothesised that in women of reproductive age, the FTE can adhere to the surface of the ovary and accumulate mutations leading to the development of ovarian inclusions, ultimately progressing to LGSOC [40]. LGSOCs harbour a high rate of activating mutations in the MAPK pathway, predominantly *KRAS* (~20%) and *BRAF* mutations (~10%), which are mutually exclusive [41,42]. *NRAS* mutations also occur in ~10% of LGSOCs [41,42].

The insulin-like growth factor 1 (IGF-1) is overexpressed in LGSOC [43]. The binding of IGF-1 to its receptor IGF-1R triggers downstream effectors such as activation of the PI3K and MAPK pathways [43]. Consistently, activating mutations in *PIK3CA* are identified in 11% of LGSOC tumours [44], and inactivating *PTEN* mutations occur in 20% of cases [45], suggesting that PI3K pathway inhibition can become a central target in LGSOCs.

In addition, mutations in *USP9X* have been reported >30% of cases between mutations and copy number loss [42]. The ubiquitin-specific protease 9X (USP9X) is an X-linked deubiquitinase involved in multiple biological processes, such as regulation of mitosis and DNA repair [46]. A recent study showed that mutations in *USP9X* increase sensitivity to mTOR inhibition, suggesting that mTOR inhibitors may represent a therapeutic approach for tumours harbouring *USP9X* mutations [46]. Similarly, mutations in the well-established OC driver gene, *ARID1A*, were identified in 9.9% of cases [42].

LGSOC exhibits resistance to conventional chemotherapy regimens, including platinum-based therapies commonly employed in OC treatment [47]. With a response rate to chemotherapy ranging from 4% to 23% [48,49], it becomes essential to develop new preclinical models to allow testing of new targeted therapies directed at classic oncogenic signalling pathways frequently active in LGSOC.

### 2.2. Type II EOCs

Type II EOCs account for 75% of all OC cases [16]. They are characterised by an aggressive nature, high prevalence of *TP53* mutations, and late-stage presentation, necessitating surgical removal plus platinum-based chemotherapy [15]. High-grade serous carcinoma, carcinosarcomas, and undifferentiated carcinomas are all grouped together as type II EOCs based on their malignant behaviour, advanced stage of diagnosis, and poor prognosis compared to type I EOCs [4].

#### High-Grade Serous Ovarian Cancer

High-grade serous ovarian cancer (HGSOC) is the most frequently diagnosed type II OC [50]. It typically presents at an advanced disease-stage and despite the early positive response to treatments, relapse is nearly unavoidable, contributing to the low 5-year survival rate (30%) [4,50]. HGSOC is generally diagnosed when the tumour has already spread throughout the peritoneal cavity, making it difficult to identify the site of origin [51]; FTE and OSE have both been implicated in the development of HGSOC [52,53].

In 2011, the Cancer Genome Atlas (TCGA) reported that somatic mutations in *TP53* occur in nearly all HGSOC (96%) [54]. Around 80% of *TP53* alterations were classified as missense mutations situated in the DNA binding domain [55,56], likely leading to a loss of function of p53 transcriptional activity [55].

The TCGA further revealed that the genomic landscape of HGSOC is characterized by profound genomic instability due to significant deficits in DNA repair pathways [54]. *BRCA1* mutations were identified in approximately 11.7% of patients, with 8.5% being germline mutations and 3.2% being somatic mutations. *BRCA1* inactivation also occurred via promoter hypermethylation in 11.5% of cases. Similarly, *BRCA2* mutations were found in about 9.2% of patients, indicating that ~30% of HGSOC harbour alterations in the *BRCA* genes [54]. However, genome instability can also be caused by acquisition of mutations affecting the double-strand break repair protein RAD50, the recombinase RAD51, and the DNA damage sensing proteins ataxia telangiectasia mutated (ATM), ataxia telangiectasia, and Rad3-related (ATR), which are all collectively mutated in 4% of HGSOCs [54]. These numbers indicate that one third of HGOSCs present defects in DNA damage repair (DDR) pathways and hence are predicted to better respond to synthetic lethal therapeutic approaches based on poly (ADP-ribose) polymerase (PARP) inhibition.

Despite the high degree of genome instability and high level of tumour mutational burden (TMB) generally predicting sensitivity to immunotherapies, OCs are considered “cold” tumours and their response to immunotherapies have been disappointing [57]. However, recent work from Marks et al. has highlighted a new and important tumour suppressive role for interferon-ε (INFε) in HGSOC [58]. The authors found that INFε is endogenously expressed in the FTE, where HGSOC originates, but generally lost in tumour cells. Importantly, using preclinical models, they showed that treatments with INFε effectively suppressed tumour cells growth and reshaped the tumour microenvironment by favouring recruitment of cytotoxic T cells and suppression of myeloid cells and regulatory T cells (Tregs) [58]. Collectively, the study proposed INFε as a potent tumour suppressor with immune properties in HGSOC [58].

The PI3K and RAS pathways are deregulated in 45% of HGSOCs [54] mainly through copy number alterations of *PIK3CA* (17%), *KRAS* (11%), *AKT1* and *AKT2* (3% and 6%, respectively), and *PTEN* deletion (7%) [54]. Given that the PI3K pathway is activated in ~70% of OC cases, several trials have been initiated targeting core kinases such as PI3K, AKT, and mTOR in combination with PARP inhibitors (PARPi) and bevacizumab [59]. Although further investigation is needed, preliminary findings suggest synergy between compounds [60].

Alterations in components of the cell cycle machinery are also frequently reported in HGSOCs [54]. The *retinoblastoma 1* (*Rb1*) gene is deleted in 8% and mutated in 2% of HGSOCs, while amplification of the *cyclin E1* gene, *CCNE1*, one of the most common focal copy number change events in HGSOC, occurs in 20% of tumours [54]. *CDKN2A*, a negative regulator of cyclins and cyclin-dependent kinases, is altered in 32% of tumour cases [54]. Collectively, these alterations have been observed in 67% of HGSOCs, indicating an important role for the retinoblastoma pathway in HGSOC treatment. Notably, loss of *Rb1* has been showed to cooperate with *PTEN* loss to initiate retinoblastoma tumours in mice [61]. This suggests that inhibition of PI3K pathway may represent a therapeutic target for tumours harbouring *Rb1* mutations [62].

NOTCH and forkhead box protein M1 (FOXM1) signalling are also involved in HGSOC pathophysiology. The TCGA reported that the NOTCH signalling pathway is deregulated in 22% of HGSOC cases [54] and elevated *neurogenic locus notch homolog protein 1* (*Notch3*) mRNA levels were observed in 63% of cases relative to benign tumours [63]. High Notch3 mRNA and protein levels also correlated with chemoresistance and poor overall survival [63]. The FOXM1 transcription factor network is activated in 84% of HGSOCs [54]. Target genes downstream of FOXM1 include *Aurora Kinase B* (*AURKB*), *cyclin B1* (*CCNB1*) *polo-like kinase 1* (*PLK1*), and target genes involved in DNA repair such as *BRCA2* and *RAD51*, which are consistently altered in HGSOC [54]. Studies have shown that upon DNA damage, p53 represses *FoxM1* mRNA, decreasing FOXM1 protein levels [64]. This may suggest that the high frequency of loss-of-function *TP53* mutations in HGSOC can contribute to the upregulation of FOXM1 and its target genes in this subtype of OC [54].

Finally, carcinosarcomas and undifferentiated carcinomas are two type II OC subtypes which are rarely diagnosed [65]. Given the limited availability of suitable samples, little is known in terms of underlying oncogenic mutations and histological features characteristic of this subtype of OC, calling for more resources to study these rare diseases.

## 3. Treatments

### 3.1. Primary Treatments

For patients diagnosed with localized, stage I OC, surgery alone is often considered sufficient to eradicate the disease, and chemotherapy may not be considered necessary [4]. However, for advanced-stage EOCs, surgical removal of the tumour followed by systemic chemotherapy is the current standard of care [4,50]. Surprisingly, the same chemotherapy regimens is administered to all OC patients, irrespective of tumour subtype [66].

In the 1970s, alkylating agents such as cyclophosphamide and anthracyclines like doxorubicin, were commonly used for the treatment of advanced OC [67]. Platinum compounds gained prominence in late 1970s, with cisplatin being the first approved drug for OC treatment. However, cisplatin was associated with significant toxicity [67] and carboplatin, introduced in the late 1980s, became a more favourable treatment [68,69]. In 1996, a key clinical trial demonstrated that taxanes like paclitaxel, an inhibitor of tubulin depolymerization [67], in combination with cisplatin outperformed cisplatin–cyclophosphamide, establishing this as the standard treatment for OC patients for over two decades [70]. Nowadays, standard chemotherapy involves intravenous (IV) carboplatin and paclitaxel (Figure 3) [71].

In OC, while primary debulking surgery followed by adjuvant chemotherapy is commonly preferred, neoadjuvant chemotherapy is also possible [73]. The choice between the two approaches defines a critical decision moment that is tailored to each individual and is based on factors including the age of the patient, extent of the disease, and existing comorbidities [74,75]. It is still not known which approach provides more benefits, and the TRUST clinical trial (NCT02828618) was purposely launched in 2016 to assess and compare the effectiveness of these two protocols.

Even though surgery and platinum-based treatments generate an initial positive response, disease relapse occurs in around 60–70% of cases [76]. Furthermore, nearly 20–30% of OC patients show overt platinum resistance [77]. The main therapeutic strategy for platinum-resistant patients with recurrent OC is systemic chemotherapy that includes PEGylated liposomal doxorubicin (PLD), paclitaxel, gemcitabine, or topotecan [71]. Bevacizumab, an angiogenesis inhibitor, can also be included for those receiving PLD, paclitaxel, or topotecan [71]. The AURELIA study demonstrated a significant 15.5% increase in response rates when incorporating bevacizumab to previous monotherapy [78].

Given the frequent platinum resistance observed in OC, novel therapeutic strategies focusing on oncogene-based targeted therapies have also been developed (Figure 4).

### 3.2. Signalling Pathways and Targeted therapies in Ovarian Cancer

#### 3.2.1. DNA Damage Repair Pathways

DNA damage repair pathways play a crucial role in maintaining genomic integrity, and disruptions of these pathways contribute to the development and progression of OC [79]. Broadly, damage to the DNA occurs through either single-strand (SSB) or double-strand breaks (DSB). Mammalian cells employ five mechanisms to detect and correct DNA damage: mismatch repair (MMR), base excision repair (BER), and nucleotide excision repair (NER) are used to fix single-strand breaks, while homologous recombination (HR) and non-homologous end joining (NHEJ) are utilized for the repair of double-strand breaks [80]. Importantly, DSB repair-mechanisms, the HR pathway in particular, are altered in up to 51% of all OCs [54] mainly by acquisition of somatic and germline mutations in *BRCA1* and *BRCA2*, which occur in 21% of HGSOCs [54].

In normal cells, the HR and BER pathways control the extent of DNA damage and compensate for each other deficiencies. In BRCA-deficient cells, the HR pathway becomes ineffective and DNA damage is repaired mainly through the BER pathway, which is coordinated by the PARP enzymes [81]. In BRCA-mutated cells treated with PARPi, the combined inhibition of both HR and BER pathways overwhelms the ability of cells to repair DNA damage causing synthetic lethality [81]. The use of PARPi has shown encouraging results in the treatment of OC, especially in patients with mutations in the *BRCA1* or *BRCA2* genes, or alterations in other DNA repair–related genes such as *partner and localizer of BRCA2* (*PALB2*), *RAD51*, *Fanconi anemia complementation group* (*FANC*), *ATM*, *ATR*, and *PTEN* [81,82].

Olaparib was the first PARPi to be approved by the U.S. Food and Drug Administration (FDA) as first-line monotherapy for advanced EOC in patients carrying germline *BRCA1* or *BRCA2* mutations who received previous lines of chemotherapy [83,84]. In 2018, the SOLO1 trial found that after a median follow-up of 41 months, the risk of disease progression, or death, was 70% lower in olaparib-treated versus placebo-treated patients [84]. This study also highlighted the therapeutic value in using PARPi not only for patients carrying *BRCA* mutations, but also for *BRCA* wildtype patients.

In 2020, the FDA approved a second PARPi, niraparib, for the treatment of patients who had complete or partial response to platinum-based chemotherapy irrespective of the HRD status. The NOVA study found that the progression-free survival (PFS) was significantly longer in the niraparib cohort compared to those receiving placebo, with or without BRCA mutations [85]. Moreover, rucaparib, a third FDA-approved PARPi, demonstrated notable advantages in maintenance therapy, following a favourable response to platinum-based chemotherapy upon recurrence [86] (Figure 3).

Beyond PARPi, possibilities exist for targeting alternative HR proteins including RAD51 and the DNA damage response kinases ATM and ATR [79]. Both ATM and ATR phosphorylate BRCA1 and p53 in order to transmit damage signals and induce a repair response [79]. Two potent and selective ATR inhibitors (ATRi), ceralasertib and berzosertib, are currently undergoing clinical evaluation. Berzosertib combined with gemcitabine exhibited extended PFS in platinum-resistant HGSOC patients compared to gemcitabine alone [87], although no improvements in overall survival (OS) were observed [88]. Ceralasertib plus olaparib did not improve objective response in a study involving 12 patients with recurrent HGSOC [89], but showed promising clinical efficacy in patients with relapsed CCC [90]. Therefore, ATR inhibitors may still represent a promising therapeutic strategy for individuals with HGSOC, and potentially also for those with less common OC subtypes.

*TP53* is a tumour suppressor mutated in nearly all HGSOCs, and in ~60% of MOC patients [34,35,54]. Loss-of-function p53 mutations result in loss of cell cycle arrest, loss of apoptosis, and chromosomal instability [91]. Given the high frequency of mutated p53 in OC, the idea of delivering a wild-type version of p53 in tumour cells with *TP53* mutations has always received a lot of interest [91]. An initial gene therapy approach using replication-deficient adenoviral vectors demonstrated initial response in patients with recurrent OC with aberrant or mutated p53, but ultimately failed to treat OC [92,93]. A second generation of p53 gene therapies is currently being evaluated. To overcome the dominant negative inhibition of mutant p53, Lu et al. developed a chimeric p53-Bad gene fusion combining p53 with the mitochondrial pro-apoptotic factor Bad [94]. The p53-Bad fusion induced apoptosis in multiple OC cell lines regardless of p53 status [94], indicating that the development of p53-based gene therapy, coupled with an optimal delivery vehicle and treatment regimen, remains a potential strategy for OC treatment.

A further way to block HR is to inhibit cyclin-dependent kinase 1 (CDK1), which is an important cell cycle regulator that participates in the HR by activating BRCA1 [95]. Preclinical studies have shown that targeting CDK1 alone with RO-3306 generates an anti-proliferative and anti-metastatic effects in multiple OC cells, and in a xenograft model of HGSOC (i.e., OSE cells) where *BRCA1* and *TP53* were specifically deleted [96]. Additionally, combined inhibition of CDK1 and PARPi in *BRCA*-proficient cells led to synthetic lethality [97]. Similarly, a combination of PARPi with WEE1 inhibitors has exhibited encouraging results in a recent phase II study [98]. WEE1 is a kinase that regulates the cell cycle by inhibiting CDK1 in response to DNA damage, thereby preventing DNA replication [99]. When WEE1 is inhibited, the cell cycle progresses without checkpoints, leading to DNA damage accumulation and cell death [99]. The WEE1 inhibitor adavosertib was evaluated alone or in combination with olaparib in 80 patients with PARPi-resistant OC [98]. The overall response rate (ORR) was 23% for adavosertib alone and 29% for patients receiving combination therapy, with a clinical benefit rate of 63% and 89%, respectively [98]. The PFS for the combination treatment was 6.8 months compared to 5.5 months for adavosertib alone [98]. However, the treatment was poorly tolerated, causing high grade toxicities [98]. To mitigate this, a recent study employed tumour-targeting nanoparticles to simultaneously deliver adavosertib and olaparib in a patient-derived OC xenograft model [100]. The nanoparticles exhibited greater tumour growth inhibition compared to the free drug combination while minimizing undesired toxic side effects [100].

While extensive research has been conducted on HR in OC, the contribution of NHEJ to OC remains less understood. McCormick et al. found that NHEJ is defective in 40% of OC cell lines and primary cell cultures derived from ascites (i.e., pathologic accumulation of fluid within the peritoneal cavity), regardless of their HR status [101]. Intriguingly, NHEJ-deficient cell lines displayed resistance to rucaparib, while sensitivity to rucaparib was specifically observed in NHEJ-competent/HR-defective cultures [101]. This suggests that NHEJ may have a significant role in determining response to PARPi. Exploring combination strategies based on PARP and key players in the DDR and cell cycle machineries can have the potential to overcome some of the resistance to PARP inhibitors.

#### 3.2.2. PI3K/AKT/mTOR Pathway

The PI3K/AKT/mTOR signal transduction pathway is one of the most frequently activated cascades in cancer, including OC [59,102,103]. Under physiological conditions, PI3K is activated by many extracellular stimuli such as growth factors, cytokines, and hormones. Upon activation, PI3K generates the lipid second messenger phosphatidylinositol-3, 4, 5-triphosphate (PIP3) and activates downstream effectors such as the protein kinase B (AKT) and the mechanistic (or mammalian) target of rapamycin (mTOR), initiating a signalling cascade that promotes growth and proliferation [104]. PTEN is the lipid phosphatase that dephosphorylates PIP3 and restrains activation of PIP3 targets, like AKT and its effector proteins [105].

Intrinsic activation of the PI3K pathway through gain-of-function *PIK3CA* mutations and/or inactivation of PTEN, is frequently observed in various cancers [106] and in many OC subtypes [102]. Consistently, *PI3KCA* and *PTEN* mutations induce ovarian tumorigenesis in mice [107] and inhibition of PI3K and mTOR with PF04691502 was found to delay tumour growth in preclinical models, even though resistance eventually occurred [107]. These finding highlight the oncogenic role of PI3K pathway activation in OC and its potential as a promising therapeutic target.

Several PI3K pathway inhibitors are currently undergoing clinical evaluation for OC treatment. Buparlisib, a pan-PI3K inhibitor, has been tested in conjunction with the MEK1/2 inhibitor trametinib and has shown a promising 29% overall response rate (ORR) in patients with *KRAS*-mutant OC [108]. However, as per many other cancers, also for OC the high incidence and severity of adverse effects impeded further clinical developments of these therapies [108]. Importantly, the efficacy and tolerability of the PI3Kα inhibitor alpelisib was recently assessed in a phase I trial in combination with olaparib in platinum-resistant OC patients [60]. In a *BRCA* wild-type setting, the ORR of the combinatorial treatment was 33.3% compared to 4–5% of olaparib as monotherapy, and <5% with alpelisib alone [60]. These promising findings suggest that PI3Kα inhibitors may enhance sensitivity to PARPi, emphasizing the potential of this synergistic combination and prompting the need for additional clinical studies. Consistently, a phase III study (NCT04729387) investigating the efficacy and safety of alpelisib in combination with olaparib compared with standard chemotherapy in patients with platinum-resistant or -refractory HGSOC with no germline *BRCA* mutation is currently ongoing [109].

Targeting AKT can provide an alternative therapeutic prospect for OC patients. Initial testing of the AKT inhibitor capivasertib in patients with *PIK3CA*-mutant cancers revealed a robust clinical response and good tolerance [110]. Subsequent phase I studies have explored the combination of capivasertib with olaparib. In a first trial combining PARP and AKT inhibitors, 44.6% of patients with advanced solid tumours achieved clinical benefit, irrespective of the BRCA1/2, DDR, and PI3K pathway status [111]. A second study in patients with endometrial, ovarian, and breast cancers reported a similar clinical benefit rate of 41% [112]. Notably, resistance to this combination was associated with elevated receptor tyrosine kinase (RTK) activity and mTOR activation [112], providing molecular insights into future combinatorial treatments for resistant patients.

Finally, mTOR inhibitors have also been tested in OC but only demonstrated limited efficacy. Among the most extensively investigated mTOR inhibitors are everolimus and temsirolimus [113]. In a recent phase I trial, the combination of everolimus with the PARPi niraparib revealed substantial toxicity in advanced OC patients even at low doses, prompting the discontinuation of the study [114]. Likewise, a separate study exploring temsirolimus in women with platinum-refractory/resistant OC was terminated due to unsatisfactory efficacy outcomes [115]. Interestingly, the dual inhibition of PI3K and mTOR with serabelisib and sapanisertib, respectively, in combination with paclitaxel was well tolerated in patients with advanced HGSOC and showed sustained clinical benefits especially in patients with alterations in the PI3K pathway [116]. Thus, in an OC subtype-specific manner, inhibition of the PI3K pathway in combination therapies may be beneficial.

Given the many crosstalk mechanisms occurring between the PI3K and MAPK pathways, the simultaneous targeting of kinases within these cascades has also been tested in preclinical CCC models [117]. The PI3K inhibitor GDC0941 was tested with the mTOR inhibitor AZD8055, and the MEK1/2 inhibitor selumetinib, in CCC cell lines and patient-derived xenograft models [117]. The low-dose, triple combination of inhibitors effectively reduced kinase activity in both the PI3K and MAPK pathways, inhibited proliferation in vitro, and significantly reduced tumour growth in vivo, indicating its potential for further clinical exploration [117].

#### 3.2.3. The MAPK Pathway

The mitogen activated protein kinase (MAPK) signalling cascade regulates key cellular processes promoting cell proliferation, cell differentiation, and cell survival [118]. Activation of the MAPK pathway is triggered by RTKs upon binding to their cognate ligands, e.g., epidermal growth factor receptor (EGFR) [119]. Dysregulation of Ras/Raf/MEK/ERK pathway is associated with OC, in particular with MOC, where 65% of cases exhibit mutations in the *KRAS* Gly-12 residue [34], and LGSOC harbouring *KRAS*, *NRAS*, and *BRAF* gene mutations [41,120].

A recent meta-analysis found that several MAPK inhibitors (i.e., Ras, Raf, MEK, and ERK inhibitors) showed 13% response rates in OC patients regardless of their subtype [121]. Additionally, in LGSOC patients with a high frequency of MAPK mutations, response rates reached 27% [121], indicating a correlation with treatment response.

LGSOC patients with the *BRAF* V600E mutation exhibited complete clinical response and maintained positive outcomes across multiple studies when treated with dabrafinib (a rapidly accelerated fibrosarcoma (Raf) inhibitor) and trametinib (a MEK inhibitor) [122,123,124]. These findings originated from studies involving a very limited number of OC patients (1–2 patients/study) but were nevertheless encouraging. Different studies have suggested that upregulation of MAPK pathway can also lead to resistance to PARPi [125,126]. Sun et al. demonstrated that *RAS*-mutant cell lines were resistant to the PARPi talazoparib and that targeting MEK or ERK could reverse this resistance [125]. It was shown that sensitization of *RAS*-mutant cells to PARPi by MEK inhibitors involves downregulation of PARP1, reduction in DNA damage sensing, and impairment of HR DNA repair capacity, ultimately enhancing sensitivity to PARP inhibition [125]. Consistent with these findings, PARP and MEK/ERK inhibitors, but not BRAF inhibitors, demonstrated synergistic activity in vitro and in vivo in OC with mutant *RAS* [125]. Results from an ongoing phase I/II study have indicated that a combination of selumetinib (MEK1/2 inhibitor) and olaparib was feasible and well-tolerated in patients with RAS alterations and PARP-resistant OC patients [127]. The *RAS*-mutant OC patients displayed the best clinical response rate (69%), with 32% of patients achieving partial response. Also, PARP-resistant OC patients responded to treatment with a clinical benefit rate of 42%, and 17% positive response [127].

#### 3.2.4. NOTCH Pathway

The NOTCH signalling pathway controls embryonic development and its dysfunction has also been implicated in many cancers, including OC [128]. Aberrant NOTCH signalling is frequently observed in HGSOC [54] and elevated NOTCH activity correlates with poor prognosis, advanced disease stage, and resistance to chemotherapy [63]. In vitro studies have shown that blocking NOTCH with the γ-secretase inhibitor DAPT effectively decreased proliferation and metastatic potential of the OVCAR-3 human OC cell line [129]. However, in patients with platinum-resistant OC, the γ-secretase inhibitor RO4929097 exhibited insufficient activity as a single-agent [130]. Given that this signalling cascade also regulates angiogenesis [131], pharmacological targeting of the NOTCH pathway in combination with anti-VEGF treatments was considered as a new potential approach. Dual inhibition of the delta-like ligand 4 (Dll4), one of the Notch ligands, with REGN1035, and VEGF with aflibercept, showed superior antitumor effects in orthotopic OC mouse models compared with monotherapy [132], confirming the therapeutic potential of this approach.

#### 3.2.5. FOXM1 Transcription Factor

The TCGA has highlighted how an activated FOXM1 transcriptional network is associated with HGSOC [54]. FOXM1 activation demonstrated strong oncogenic properties in regulating cell proliferation and cell cycle progression, as well as formation of metastasis in human cancer cells [133]. Therefore, targeting FOXM1 could potentially disrupt multiple oncogenic pathways simultaneously. Recently, Zhang et al. showed that XST-20 effectively suppressed transcriptional activities of FOXM1, leading to a significant reduction in colony-forming efficiency, increased cell cycle arrest and apoptosis across multiple OC cell lines [134]. However, as with other signalling pathways, FOXM1 inhibition also results in the activation of compensatory signalling pathways, leading to treatment inefficacy and drug resistance [135]. In order to overcome this adaptive response, a combination of FOXM1 inhibitors with other therapeutic agents such as tipifarnib (NRAS inhibitor) [135] and olaparib [136] has been tested and exhibited promising results. However, additional in vivo studies are necessary for future clinical application.

#### 3.2.6. Angiogenesis

Angiogenesis refers to the formation of new blood vessels from pre-existing vasculature [137]. Angiogenesis is a key hallmark of cancer and plays a pivotal role in OC progression by promoting tumour growth and metastasis [137]. Vascular endothelial growth factor (VEGF) is a potent pro-angiogenic growth factor that promotes vascularity in response to hypoxic conditions, an essential environmental factor influencing correct ovarian follicle development [137,138]. Notably, high levels of VEGF in ascites significantly contributes to peritoneal metastases in OC. High VEGF levels in the abdomen are associated with platinum resistance and poor prognosis [139], indicating that there is value in targeting VEGF in OC.

Two angiogenesis inhibitors targeting the VEGF receptor, cediranib, and circulating VEGF, bevacizumab, have demonstrated anti-tumour activity when used individually, with 21% response rates in patients with advanced OC [140,141]. Combining bevacizumab with chemotherapy, followed by its use as maintenance therapy, has generated improved PFS for both newly diagnosed and recurrent OC [142,143,144]. These findings led to the addition of bevacizumab to paclitaxel and carboplatin every 3 weeks as standard-of-care in advanced OC patients in multiple countries [145] (Figure 3).

To date, combinations of PARPi with anti-angiogenic therapies stand out as some of the most extensively studied combinatorial treatments in OC [146]. While the underlying mechanisms for the effectiveness of these treatments are not fully understood, previous research has proposed that hypoxic stress caused by anti-VEGF therapy (i.e., bevacizumab) [139] can induce suppression of the HR pathway [147] and sensitizes OC cells to PARPi. In line with this, a combination of niraparib plus bevacizumab significantly improved progression-free survival compared with niraparib alone (11.9 months vs. 5.5 months), irrespective of the HR status [148]. Although combination therapies of PARP inhibition with anti-angiogenic agents have not been demonstrated to be superior in terms of OS and hazard ratio to standard-of-care chemotherapy [149], additional research is required to explore molecular biomarkers that identify patients most likely to benefit from this combination.

#### 3.2.7. Cancer Immunotherapy

Immunotherapies exploit the immune system to identify and attack malignant tumour cells, and have revolutionised our approach to treat cancer, including for OC [150]. Immunotherapy based on immune checkpoint inhibitors (ICIs) targeting cytotoxic T lymphocyte antigen 4 (CTLA-4) [151], or the programmed death 1 (PD-1) receptor [152,153,154], have been tested alone or in combination with anti-angiogenic agents and PARPi in EOC. In combination therapies, niraparib plus the anti-PD-1 antibody pembrolizumab has shown promising efficacy in recurrent platinum-resistant OC, with 36% of patients achieving partial response and 50% experiencing stable disease [155]. A recent phase II clinical trial combining bevacizumab, olaparib, and durvalumab (anti-PD-1 therapy) exhibited an encouraging degree of efficacy in patients with platinum-resistant OC [156]. Furthermore, dual checkpoint inhibition with ipilimumab (anti-CTLA-4 therapy) and nivolumab (anti-PD-1 therapy) demonstrated increased objective response rate (31.4% vs. 12.2%) and progression-free survival (3.9 months vs. 2 months) compared with nivolumab alone [157].

The presence of intraepithelial CD8^+^ cytotoxic tumour-infiltrating lymphocytes (TILs) has been associated with improved survival in OC and better response to ICIs; however, OCs tend to show limited TIL infiltration [158]. As an alternative, T-cell transfer therapy or adoptive T-cell therapy (ACT) has recently garnered considerable attention for OC treatment. In a case study, Pedersen et al. showed that ACT for metastatic OC was well tolerated and showed initial clinical activity [159]. However, increased PD-1/PD-L1 and LAG3/MHCII checkpoint pathways inhibited T cell activation, led to an exhausted T cell phenotype, and caused transient clinical response [159]. One way to potentiate the clinical efficacy of TIL therapy in OC could be to combine treatments with checkpoint inhibitors. In a study led by Kverneland et al., six patients with late-stage metastatic HGSOC received a combination therapy consisting of ipilimumab and nivolumab [160]. Promising results were observed, with one patient achieving partial response and five others experiencing disease stabilization for up to 12 months [160]. Optimization of immunotherapies can also be tested in immunocompetent murine models based on the epithelial ID8 OC cell line, as previously reviewed in Rodriguez et al. [161].

#### 3.2.8. Endocrine Therapy

OCs commonly express steroid hormone receptors such as the estrogen receptor (ER), which is variably expressed in 25–86% of OC cases [162,163]. Hormonal therapies have been considered as a potential treatment for patients with ER+, advanced OC, given the high efficacy achieved with ER+ breast cancers. However, not all OC cancers positive for ER respond to anti-estrogens. In fact, the percentage of ER+ OC cases showing positive response to anti-estrogen therapy such as tamoxifen is relatively low (10–15%) compared to the ~ 80% response rates observed in ER+ breast cancer patients [164,165]. What drives these different outcomes between ovarian and breast cancer patients is not completely understood. It is possible that the underlying genetic mutations causing OC effectively overcome the benefits of ER inhibition.

Nevertheless, a meta-analysis of 53 clinical trials involving 2,490 EOC patients revealed a collective 41% clinical benefit for endocrine regimens with specific rates of 43% for treatments based on tamoxifen, 39% for aromatase inhibitors, and 37% for progestins [166]. Notably, endocrine therapies reduced mortality rates [166]. Consistent with this, a retrospective study by Gershenson et al. showed an improved PFS in LGSOC patients treated with hormonal maintenance therapy compared to patients with non-hormonal treatment after cytoreductive surgery and platinum-based chemotherapy (64.9 months vs. 26.4 months) [167].

Collectively, these results provide a strong rationale for the therapeutic use of endocrine therapies for hormone receptor-positive OC, particularly given their tolerability, low cost, and promising efficacy. A phase III trial is currently underway to also evaluate the inclusion of aromatase inhibitors such as letrozole to the standard maintenance therapy for ER+ EOC patients [168].

## 4. Experimental Models of Ovarian Cancer

Conventional two-dimensional (2D) cell lines derived from OC patients have provided an invaluable model system to study various aspects of the disease, from its molecular underpinnings to therapeutic targeting [169]. However, the generation of 2D cell lines derived from tissue biopsies has been characterized by limited and unpredictable success rates [170]; in addition, monocultures tend to over-simplify the genetic heterogeneity exhibited by tumours in vivo (Figure 5) [171].

Several 2D EOC cell lines commonly used worldwide were established decades ago, when cell line validation was not fully implemented. As an example, Domcke et al. found that two of the most frequently used OC cell lines, SKOV-3 and A2780 cells, were not molecularly representative of HGSOC, despite been initially labelled as such [172]. Similarly, the IGROV-1 cells were classified as EnOC rather than HGSOC, inaccurately labelled in the literature [172]. Advances in genome sequencing has enabled a comprehensive analysis of existing OC cell lines, aiding the detailed characterization of their histological and molecular features [173,174]. Improved culture conditions have also allowed for the successful creation and maintenance of new OC cell lines, expanding the representation of various ovarian tumour subtypes [175].

As for many other cancer types [176], the derivation of three-dimensional (3D) organoid lines from patients’ biopsies has become a common practice also in OC research. Organoids are in vitro 3D stem cell-like cultures derived from healthy donors or tumour samples [177]. Importantly, a percentage of tumour organoids has been shown to retain most of the genetic and phenotypic functions of the primary tissues [178], rendering them a promising platform for screening anticancer drugs.

To establish patient-derived organoids (PDOs) from surgically removed OC biopsies, purified cells are embedded in a 3D extracellular matrix scaffold, commonly Matrigel, and cultured in a nutrient-rich medium, supplemented with cocktails of growth factors and hormones to ensure their long-term maintenance [16]. Numerous protocols have been developed to establish OC organoids [178,179,180,181,182]. In 2019, Kopper and colleagues introduced a robust protocol enabling long-term cultures of organoids derived from MOC, CCC, EnOC, LGSOC, and HGSOC, with an overall success rate of 65% [179]. These PDOs preserved the genetic profile, histological characteristics, and heterogeneity of the original tumour, even after prolonged in vitro passaging [179]. Similarly, Maenhoudt et al. successfully generated organoids from HGSOC biopsies, which faithfully reproduced the molecular and cellular phenotype of the primary tumours [183]. Notably, this study identified neuregulin-1 as a key factor for the development and growth of OC organoids [183]. However, subsequent research reported that neuregulin-1 could stimulate but also restrict organoid growth, depending on the sample and its origins [178].

Organoids provide an important ex vivo model to test patient-specific response to drugs as they often recapitulate characteristics of the primary tumour [178,184,185]. Recently, Senkowski et al. found that response of HGSOC organoid to drug treatments correlated with clinical outcomes. Importantly, the culture medium influenced the drug response rates as it was found that supplements of N-acetylcysteine influenced sensitivity of cancer cells to platinum drugs [178,186]. Nonetheless, tumour organoids remain an important predictive model for personalized medicine, particularly if grown in co-culture settings. Malacrida et al. have recently described the generation of tri- and tetra-cultures involving HGSOC organoids [187]. In this study, omental tissue from patients was collected to extract mesothelial cells, fibroblasts, and adipocytes. These cells were successfully co-cultured with tumour cells plated within an adipocyte gel medium [187], showing that the reproduction of complex tumour microenvironment (TME) in vitro is achievable.

Patient-derived xenografts (PDX) offer a more complete representation of the TME compared to in vitro co-cultures and can retain the original tumour histology and molecular profile, along with similar patient’s response to therapy [188,189]. Multiple PDXs models have been established for different subtypes of EOC [189,190,191,192]. Cybula et al. successfully engrafted 33 primary HGSOC into immunodeficient mice with a 77% engraftment rate [189] and reported that tumour engraftment was significantly associated with early tumour recurrence (within 12 months since diagnosis), rather than factors such as patient age, tumour stage, or response to platinum-based chemotherapy and OS [189]. PDXs have also been established from MOC, EnOC, and CCC, although with varying engraftment rates: 30%, 60%, and 80%, respectively [192].

Given the ability of PDXs to preserve some of the features of the human TME, PDXs models of OC have been used for preclinical drug evaluation and biomarker identification [193]. As an example, Harris et al. established and characterized a collection of HER2-positive HGSOC PDXs and found that while HER2-targeted therapy resulted in modest tumour inhibition, a combination therapy including chemotherapy plus HER2 inhibitors improved treatment response [194]. This highlights how, despite the high-costs and variable engraftments rates [195], PDXs may define some of the most advanced co-clinical model systems to study OC response to treatments.

## 5. Conclusions

Ovarian cancer is a lethal gynaecological malignancy that presents with challenges in detection and resistance to treatments. Our current understanding of the molecular features characterising ovarian cancer has progressed dramatically in recent years, and new oncogene-based targeted therapies have been considered as alternative treatments to standard systemic therapies. However, ovarian cancer comprises a collection of diseases, and several tumour subtypes such as the mucinous remain poorly characterised. The increasing appreciation of the contribution of various signalling pathways to OC survival and growth, such as the PI3K, MAPK, and WNT pathways, is emerging as a new avenue for the identification of new and more effective combination therapies to overcome chemo-resistance. Thus, in the era of personalised medicine, treatment models informed by new biomarkers, and based on combinatorial treatments, can become a vital approach to undercut the molecular and genetic complexity of ovarian cancer.

## Figures and Tables

**Figure 1 biomolecules-14-00585-f001:**
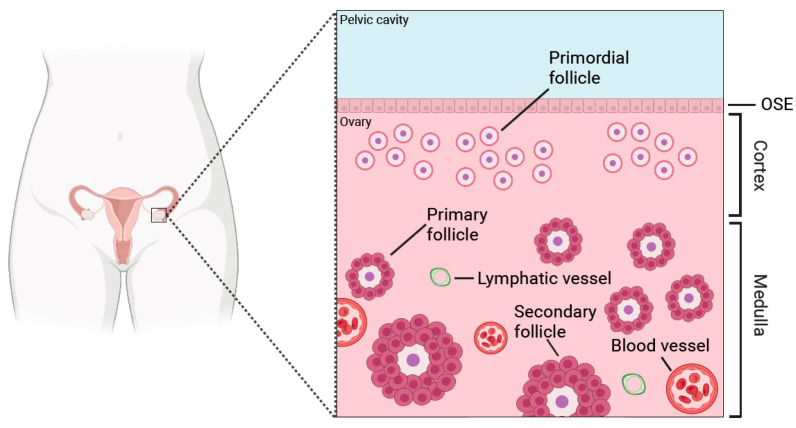
Cartoon depicting a cross section of the human ovary. The cortex, the outer part of the ovary, harbours quiescent primordial follicles that define the ovarian reserve. Beneath the cortex, the medulla and its connective tissue contain abundant blood and lymphatic vessels plus growing follicles; primary and secondary follicles are illustrated. OSE: ovarian surface epithelium.

**Figure 2 biomolecules-14-00585-f002:**
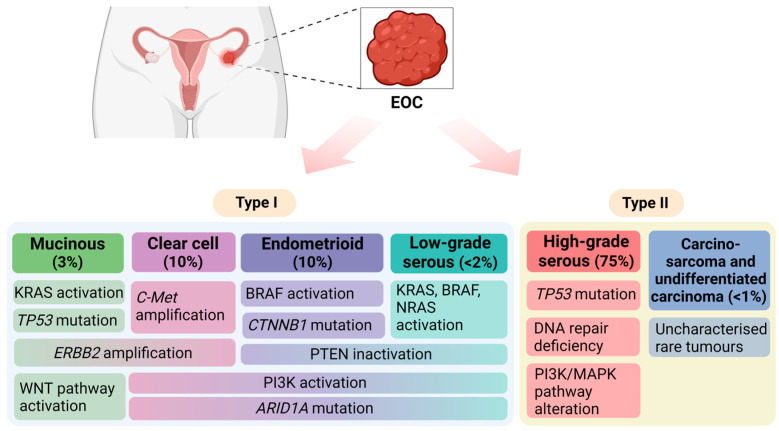
Epithelial ovarian cancer (EOC) classification. Type I carcinomas comprise: (1) mucinous carcinoma (MOC); (2) clear cell carcinoma (CCC); (3) endometrioid ovarian cancer (EnOC); and (4) low-grade serous ovarian cancer (LGSOC). Type II carcinomas are largely composed of high-grade serous ovarian cancer (HGSOC), carcino-sarcoma and undifferentiated carcinoma. Frequently reported gene mutations and associated signalling pathways are included for each EOC subtype. ARID1A: AT-rich interaction domain 1A; BRAF: a serine/threonine-specific kinase; *c-Met*: c-mesenchymal–epithelial transition factor; *CTNNB1*: β-catenin; *ERBB2*: estrogen-related receptor β2; KRAS and NRAS: members of a specific GTPase superfamily; PI3K: phosphatidylinositol-4,5-bisphosphate 3-kinase; PTEN: phosphatase and tensin homolog deleted on Chromosome 10; *TP53*: tumour protein p53.

**Figure 3 biomolecules-14-00585-f003:**
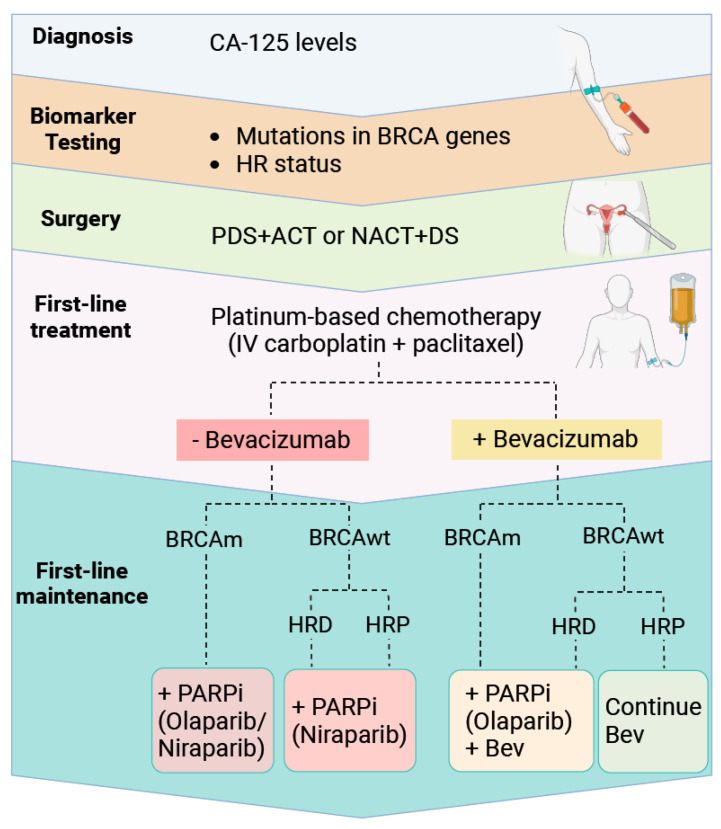
Standard protocols for ovarian cancer treatment. After diagnosis, patients undergo blood tests to detect BRCA mutations and monitor the status of homologous recombination (HR) pathways. Following consultation, surgeons determine the timing of debulking surgery in relation to chemotherapy. Standard-of-care treatments for ovarian cancer involves platinum-based chemotherapy with or without bevacizumab, an angiogenesis inhibitor. Depending on genetic results, maintenance therapy may include PARP inhibitors (PARPi), bevacizumab, or a combination of both [72]. ACT: adjuvant chemotherapy; BRCA: breast cancer gene; BRCAm: breast cancer gene mutant; BRCAwt: breast cancer gene wildtype; Bev: Bevacizumab; DS: debulking surgery; HRD: homologous recombination deficiency; HRP: homologous recombination proficiency; IV: intravenous; NACT: neoadjuvant chemotherapy; PARP: poly (ADP-ribose) polymerase; PDS: primary debulking surgery.

**Figure 4 biomolecules-14-00585-f004:**
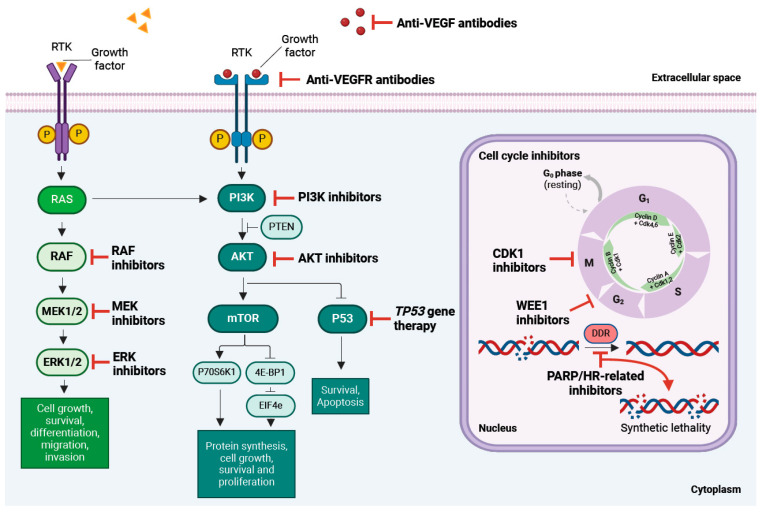
Molecular targets and signalling pathways currently under investigation for ovarian cancer treatment. 4E-BP1: eukaryotic translation initiation factor 4E-binding protein 1; AKT: protein kinase B; CDK1: cyclin dependent kinase 1; DDR, DNA damage repair; ElF4e, eukaryotic translation initiation factor 4E; ERK: extracellular signal-related kinase; HER2: human epidermal growth factor receptor 2; HR, homologous recombination; RAS: member of a specific GTPase superfamily; MEK: MAPK/ERK kinase; mTOR, mechanistic (or mammalian) target of rapamycin; PARP, poly (ADP-ribose) polymerase; P53, tumour protein p53; PI3K: phosphoinositol 3-kinase; PTEN, phosphatase and tensin homolog deleted on Chromosome 10; RAF: a serine/threonine-specific kinase; RTK, receptor tyrosine kinase; S6K1: S6 kinase beta-1; VEGF: vascular endothelial growth factor; VEGFR: vascular endothelial growth factor receptor; WEE1: nuclear tyrosine kinase.

**Figure 5 biomolecules-14-00585-f005:**
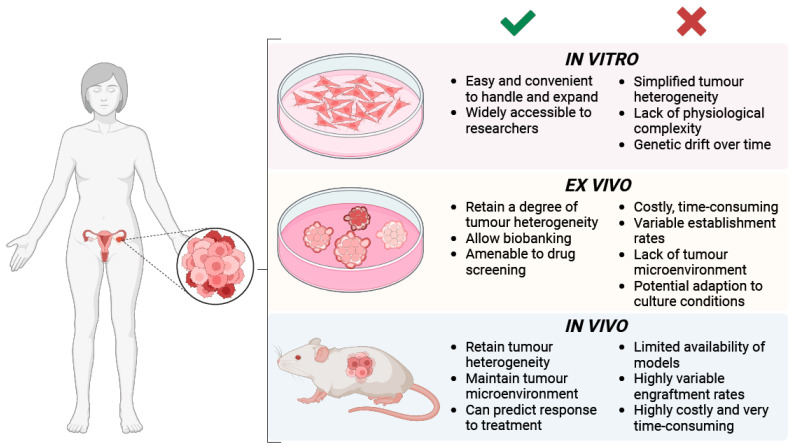
Experimental models currently used to study and treat ovarian cancer including immortalised cell lines, patient-derived organoids (PDOs), and patient-derived xenografts (PDXs).

## Data Availability

No new data were included in this review.

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
