# Peer review of "Oncogenic Pathways and Targeted Therapies in Ovarian Cancer"

_biomolecules, 2024, doi:10.3390/biom14050585_

Round 1

Reviewer 1 Report

Comments and Suggestions for Authors

This is a very well-written and comprehensive review. That said, more could be done with immunotherapy. Clinical trial are being done on PD-1 inhibition for ovarian clear cell carcinoma. In addition, adoptive T cell therapy as it applies to ovarian cancer is worth discussing. 

In terms of models, it could be mentioned that ID8 cells injected into C56Bl/6 mice represents an immunocompetent preclinical model. 

Author Response

A response to the Reviewer's comments has been uploaded.

Reviewer 2 Report

Comments and Suggestions for Authors

This manuscript is comprehensive and well-organized review about ovarian cancer with genetic landscape, current targeted therapies and preclinical studies for the future treatments.  I have no major concerns.  Here are minor issues need to be addressed.

1) Line 468: remove "apoptosis"

2) Line 474: It is unclear which "MAPK inhibitors" the authors meant.  Please define.

3) Line: 480: MEK kinase ---> MEK

4) Some reference numbers in the text are not matched with citations.  Please check this issue.

Author Response

(The authors gave the same response as above.)

Reviewer 3 Report

Comments and Suggestions for Authors

Ovarian malignant tumor is one of the common malignant tumors of female reproductive organs, and its incidence rate is only second to cervical cancer and endometrial cancer. This article provides a detailed review of the research progress and some treatment strategies of Epithelial ovarian cancer. It is a good reference and learning for researchers in this field. I have some questions. Type II tumors are high-grade serous and carcinosarcoma, containing mutations in the p53, BRCA1, and BRCA2 genes. Meanwhile, the NOTCH and FOXM1 signaling pathways are associated with the pathological and physiological mechanisms of ovarian serous carcinoma. The author can appropriately introduce some new treatment strategies, endocrine therapy and hormone replacement therapy. A study has found that approximately 60% of ovarian cancer samples detect estrogen receptors, but the disease is not sensitive to estrogen. Endocrine drugs, such as tamoxifen or letrozole, are occasionally effective. The hormone replacement therapy for gynecological malignant tumor patients is the second important issue. Due to young patients under the age of 50 being exposed to estrogen, hormone replacement therapy is generally safe. PI3K and MAPK have a wide range of pathways of action, and are they suitable as targeted drugs in clinical practice.

Comments on the Quality of English Language

Moderate editing of English language required

Author Response

(The authors gave the same response as above.)

Reviewer 4 Report

Comments and Suggestions for Authors

This review manuscript is well written. It's suitable for publication in Biomolecules.

Author Response

(The authors gave the same response as above.)
